# Marine Phytoplankton Bioactive Lipids and Their Perspectives in Clinical Inflammation

**DOI:** 10.3390/md23020086

**Published:** 2025-02-17

**Authors:** Edoardo Andrea Cutolo, Rosanna Campitiello, Valeria Di Dato, Ida Orefice, Max Angstenberger, Maurizio Cutolo

**Affiliations:** 1Laboratory of Photosynthesis and Bioenergy, Department of Biotechnology, University of Verona, Strada le Grazie 15, 37134 Verona, Italy; 2Laboratory of Experimental Rheumatology and Academic Division of Clinical Rheumatology, Department of Internal Medicine, University of Genoa, 16132 Genova, Italy; rosannacampitiello@hotmail.it (R.C.); mcutolo@unige.it (M.C.); 3IRCCS Ospedale Policlinico San Martino, 16132 Genova, Italy; 4Stazione Zoologica Anton Dohrn Napoli, Ecosustainable Marine Biotechnology Department, Via Ammiraglio Ferdinando Acton 55, 80133 Napoli, Italy; valeria.didato@szn.it (V.D.D.);; 5Institute of Molecular Biosciences, Goethe University Frankfurt am Main, Max-von-Laue-Str. 9, 60438 Frankfurt am Main, Germany; m.angstenberger@bio.uni-frankfurt.de

**Keywords:** microalgae, bioactive lipids, eicosanoids, inflammation, diatoms

## Abstract

Marine phytoplankton is an emerging source of immunomodulatory bioactive lipids (BLs). Under physiological growth conditions and upon stress challenges, several eukaryotic microalgal species accumulate lipid metabolites that resemble the precursors of animal mediators of inflammation: eicosanoids and prostaglandins. Therefore, marine phytoplankton could serve as a biotechnological platform to produce functional BLs with therapeutic applications in the management of chronic inflammatory diseases and other clinical conditions. However, to be commercially competitive, the lipidic precursor yields should be enhanced. Beside tailoring the cultivation of native producers, genetic engineering is a feasible strategy to accrue the production of lipid metabolites and to introduce heterologous biosynthetic pathways in microalgal hosts. Here, we present the state-of-the-art clinical research on immunomodulatory lipids from eukaryotic marine phytoplankton and discuss synthetic biology approaches to boost their light-driven biosynthesis.

## 1. Introduction

Non-communicable immune-mediated inflammatory diseases (IMIDs) are a major burden for healthcare systems worldwide, with a greater incidence in higher income regions [1]. Different causes underlie the development of IMIDs, including genetics and lifestyle factors [1]. Among the latter, diet plays a prominent role in mitigating the risk of disease onset and progression towards chronic inflammation. Anti-inflammatory diets typically include plant-based sources of lipids that are capable of modulating the pathways involved in the pathogenesis of chronic inflammation [2].

A hallmark of chronic IMIDs is the altered metabolism of endogenous and dietary lipids. Lipids are essential constituents of cellular membranes and the storage of metabolic energy, but are also the biosynthetic precursors of signaling molecules, or bioactive lipids (BLs), which regulate inflammatory processes [3]. Therefore, nutritional interventions can significantly influence the balance between BL species and their immunomodulatory activity [4].

Inflammation is a physiological mechanism to counteract infections and tissue damage, and its timely resolution is part of a healthy bodily response. However, a persistent pro-inflammatory immune activation is at the origin of IMIDs. Virtually all immune cell types synthetize and release lipid-based soluble mediators that coordinate the initiation, progression and resolution of inflammatory processes by regulating vascular hyperreactivity, pain, leukocyte trafficking, and clearance. The failure in confining acute inflammation promotes the transition from acute to chronic inflammation. Indeed, the development and progression of all major IMIDs (i.e., rheumatoid arthritis, atopic dermatitis, asthma, multiple sclerosis, type-1 diabetes, systemic lupus erythematosus, psoriasis, and inflammatory bowel disease), involves an imbalance between the activity of pro-inflammatory and resolving BLs [3,5]. The main classes of BLs involved in the initiation of inflammation and its progression towards chronic conditions are the eicosanoids. Other types of lipid-derived metabolites that play crucial roles in the resolution of inflammation include the resolvins (specialized pro-resolving mediators), lysoglycerophospholipids, sphingolipids, and endocannabinoids [6,7,8]. The dietary intake of marine lipids, namely, long-chain polyunsaturated fatty acids (PUFAs), including omega-3 lipids from fish, fish oils, and seafood, lowers the risk of IMIDs, ameliorates the disease activity, and reduces the need for long-term use of anti-inflammatory drugs [9,10,11,12].

The supply of marine lipids to an increasing global population through the fishery industry, however, faces several challenges with respect to the preservation of natural ecosystems [13,14,15,16]. Therefore, the discovery of sustainable sources of dietary lipids is highly desirable.

Ocean ecosystems have a long history as sources of active compounds [17,18] and are currently witnessing a scientific renaissance as troves of metabolites endowed with anti-inflammatory and immunomodulatory properties [19,20,21,22,23,24]. Eukaryotic marine phytoplankton (hereafter microalgae) is an emerging alternative to metazoans to obtain dietary BLs [25,26,27,28,29], with an increasing number of species holding Generally Recognized As Safe and Novel Food statuses [30,31,32,33]. Microalgae hold the greatest promise as producing platforms of valuable PUFAs like eicosapentaenoic acid (EPA) and docosahexaenoic acid (DHA), although other BL types have been reported [34,35,36]. The complexity of phytoplankton biodiversity and lipid chemodiversity is beginning to surface and few species have been tested in pre-clinical and clinical studies for their potential in the management of chronic inflammation [37,38]. A relatively recent discovery is the existence of biosynthetic pathways of prostaglandins (PGs) in diatoms, a class of hormone-like eicosanoids (or prostanoids) acting mainly as pro-inflammatory modulators [39,40,41,42].

Considering the growing body of literature dedicated to the characterization and emerging clinical potential of BLs from marine phytoplankton [43,44], this review aims to provide an updated summary of this rapidly developing research field [45,46]. We cover recent reports describing microalgal BLs and the emerging genetic engineering techniques to enhance their yields and to produce non-native lipids [47].

## 2. Bioactive Lipids in Human Physiology and Disease

Eicosanoids are the main class of signaling BLs regulating innate immune responses, including the initiation and resolution of inflammation. These short-lived autocrine and paracrine are produced from 20-carbon-chain PUFAs, mainly from the omega-6 arachidonic acid (AA, series 2 PGs, PGE_2_ pro-inflammatory) and, to a lesser extent, from the omega-3 EPA (series 3 PGs, e.g., PGE_3_ anti-inflammatory) [48,49] and dihomo γ-linolenic acid (20:3, *n* − 6, DGLA, series 1 PGs), the latter a derivative of linolenic acid and biosynthetic intermediate of AA [50,51,52,53,54].

Eicosanoids comprise a variety of structurally related molecules, among which are PGs, prostacyclins, thromboxanes (TXs) known as prostanoids, and leukotrienes (LTs). The synthesis of eicosanoids proceeds with the release of PUFA precursors from the phospholipid bilayer of cellular membranes by phospholypase enzymes and their oxidation by cyclooxygenases 1 and 2 (COX-1/2) in the case of prostanoids, while LTs are produced by lipoxygenases [41,55]. These products are, therefore, referred to as oxylipins. Non-enzymatic oxylipins can also originate from radical-mediated spontaneous lipid peroxidation, such as the isoprostanoids (or isoprostanes) and the less-characterized isofurans [56,57,58,59,60,61], giving rise to a wider array of BLs metabolites, many of which are endowed with anti-inflammatory, immunomodulatory, and neuroprotective properties [62,63,64,65,66,67,68,69,70,71]. Following the activation of phospholipase A2, AA is released and subsequently converted through the COX pathway into PGs [72,73,74]. The AA cascade is central to the inflammatory process as it also produces TXs, which promote platelet aggregation and vasoconstriction, and LTs, which enhance vascular permeability [55].

Of note, the two COX-1 and COX-2 isoforms display differential cell type and tissue expression and regulation during the progression of inflammatory processes in response to specific cytokines [75,76,77], highlighting the duality of PG functions [78]. While COX-1 is constitutively expressed throughout the body providing homeostatic functions [79], the inducible COX-2 isoform is strongly upregulated by inflammation and increases the local production of PGs involved in mediating inflammation and pain, although it is suggested that it might also play some homeostatic roles [80,81,82,83].

PGs were discovered in the 1930s by Swedish physiologist Ulf von Euler in human semen and originally believed to be produced by the prostate gland, hence their name [84]. The pivotal roles of PGs in inflammation were reported in the 1960s by British scientist Sir John Vane, who described their inhibitory effects of nonsteroidal anti-inflammatory drugs (NSAIDs) [85]. Vane further characterized prostacyclin, a PG-related molecule that prevents blood clotting in a delicate balance with TXs [85]. PGs regulate multiple physiological processes ranging from blood flow regulation to inflammation and immune responses, and can be responsible for disease states (summarized in Figure 1) [86].

The main physiological effects of PGs consist in the enhancement of vascular permeability, to allow immune cells access to the site of injury [87]. Notably, prostaglandin E_2_ (PGE_2_) can exert either immunomodulatory or pro-inflammatory effects by suppressing T-cell proliferation or regulating B-cell differentiation, respectively [88,89,90]. PGE_2_ also regulates vascular tone by acting as a vasodilator and plays a critical role in maintaining renal blood flow. Furthermore, PGE_2_ enhances its own biosynthesis and suppresses acute-inflammatory mediators, thus playing a prominent role in the late/chronic stages of immunity [91]. The biological effects of PGE_2_ strictly depend on the G-protein-coupled receptor (EP) to which it binds on immune cells EP1, EP2, EP3, and EP4, each displaying tissue-specific expression and being associated with different signaling pathways [92]. For instance, PGE_2_ induces fever through EP3 receptor activation in the hypothalamus and enhances pain perception by increasing neuronal excitability, while binding to EP2 and EP4 activates both inflammation and immunosuppression [87,93,94,95]. Therefore, PGE_2_ has been tested as an immunomodulatory agent and cancer-immunotherapeutic both in vitro and in vivo animal studies, although reports on clinical uses of this mediator are scarce [96,97,98,99]. Other examples of prominent PGs are prostaglandin F2α (PGF_2_α), which induces uterine smooth muscle contractions, particularly during labor, and can influence the tone of bronchial and gastrointestinal smooth muscle, often causing constriction [100]. The cell-type-specific expression of PGE_2_ receptors underscores the manifold roles played by this molecule in the maintenance of homeostasis [101,102,103,104,105,106]. While mastocytes residing in connective tissues mainly produce PGD2, macrophages synthetize PGE_2_ and TXA_2_ [74]. The dynamic modulation of PG synthesis, therefore, reflects the complex regulation of immune responses, in which PGs regulate cellular functions depending on the chemical environment [74].

Autoimmune diseases are characterized by the sustained secretion of pro-inflammatory mediators known as cytokines (mainly interleukin 17, IL-17) by T lymphocyte cells (T cells), which is otherwise restrained by counteracting cytokines (tumor necrosis factor β, TNF-β, and IL10) released by dendritic cells. The de-regulated T-cell activity is further exacerbated by leukocyte recruitment and the activation of macrophages, which produce and secrete the pro-inflammatory mediators IL-6, TNF-α, IL-17, and PGE2 [107]. PGE2 is primarily responsible for the differentiation of the highly reactive TH1 cell type and the expansion of the TH17 population [108] and, at the same time, reduces the biosynthesis of the resolving BLs lipoxins (derived from AA), resolvins (derived from EPA and DHA), protectins (derived from DHA), and maresins (derived from DHA) [109].

In clinical practice, COX inhibitors are routinely employed to mitigate PG-mediated chronic inflammation [110]; however, they non-specifically inhibit both COX1 and COX2 isoforms [111,112]. The prolonged use of NSAIDs, however, is typically associated with dangerous side effects such as gastrointestinal, cardiovascular, and renal damage [113,114]. Therefore, it is suggested that the administration of precursors of resolving BLs such as EPA and DHA could be an effective strategy to mitigate the risk of developing chronic inflammatory conditions, or to modify their course.

Due to their primary functions as pro-inflammatory mediators, PGs find relatively few clinical applications. For instance, PG analogs such as latanoprost, bimatoprost, and travoprost are employed as first-line therapies to lower intraocular pressure in glaucoma patients. These prodrugs undergo hydrolysis by corneal esterases, producing metabolites that bind to the prostaglandin F (PGF) receptor, increasing uveoscleral outflow and, thus, reducing the risk of optic nerve damage and vision loss [115,116,117,118,119]. The PG analogs dinoprostone (PGE2) and misoprostol (PGE1) are used in obstetrics to induce labor and stimulate myometrial contractions [120], particularly in cases of post-term pregnancy or preeclampsia [121].

## 3. Microalgal PUFAs, EPA, and DHA: Clinical Perspectives

Microalgal lipid biosynthesis begins with de novo production in the plastid and proceeds with extra-chloroplast biosynthesis in the endoplasmic reticulum [122]. Microalgae produce different types of PUFAs that could serve as precursors to resolving BLs [123,124,125]. Moreover, several independent studies reported that marine microalgae accumulate around 70 different types of non-enzymatic oxylipins [126,127,128,129].

Remarkably, several strains display an extreme metabolic plasticity in response to environmental stresses, including variations of light intensity, which is reflected in dramatic changes of the cellular lipidome, as recently reported for the eustigmatophyte *Nannochloropsis gaditana* [130,131]. Microalgae were already suggested two decades ago as alternative sources of dietary lipids [132], mainly of the three omega-3 PUFAs ALA (C18:3, *n* − 3), EPA (C20: 5, *n* − 3), and DHA (C22:6, *n* − 3). The immunomodulatory effects of different microalgal lipids have been tested in vitro and in vivo, indicating their potential clinical use (summarized in Figure 2). For instance, the lipid extracts enriched in EPA and DHA from the haptophyte *Pavlova lutheri* (Prymnesiophyceae) inhibited the release of the pro-inflammatory cytokines IL-6 and TNF-α by suppressing the intracellular nuclear factor kappa Β (NF-kB)-mediated pathway in cultured activated human macrophages, a major pro-inflammatory signaling cascade involved in several IMIDs [133,134]. Additionally, a more recent study reported similar effects with extracts from *Nannochloropsis oceanica* and the chlorophyte *Chlorococcum amblystomatis*, which suppressed the synthesis of the pro-inflammatory mediators nitric oxide, IL1-β,ß and TNF-α [135].

Studies in vivo have also investigated the effects of dietary supplementation with DHA and EPA from the *Nannochloropsis* sp. and the haptophyte *Isochrysis galbana*, showing changes on the plasma PUFAs levels in animal models [136]. Moreover, supplementation with DHA extracts from the haptohyte *Tisochrysis lutea* in an animal model of metabolic syndrome lowered the plasma levels of the pro-inflammatory marker TNF-α, while increasing the production of the anti-inflammatory cytokine IL-10 [137]. Of note, a recent clinical trial reported the safety profile of *Nannochloropsis*-derived DHA and EPA fed to patients with hypertriglyceridemia, highlighting a greater reduction in plasma triacylglycerols compared with supplementation with corn oil/soy oil [138]. Finally, in two recent clinical trials, the EPA-rich extract from *Nannochloropsis* commercially known as Almega^®^PL improved the Omega-3 Index and cardio-metabolic parameters and lowered the plasma cholesterol levels in healthy individuals [139,140].

Several marine strains accumulate so-called betaine lipids, glycerolipids with ether-bond charged betaine groups attached to the glycerol backbone in place of the phosphate and carbohydrate moieties endowed with anti-inflammatory and immunomodulatory properties [141,142]. Betaine lipids are produced in microalgae upon membrane remodeling in response to abiotic stresses, such as nitrogen starvation, and act as acyl group donors for de novo lipid biosynthesis [143,144,145,146,147]. The most abundant class of betaine lipids found in microalgae is 1,2-diacylglyceryl-3-O-4′-(N,N,N-trimethyl)-homoserine (DGTS), followed by 1,2-diacylglyceryl-3-O-carboxy-(hydroxymethyl)-choline (DGCC) and 1,2-diacylglyceryl-3-O-2′-(hydroxymethyl)-(N,N,N-trimethyl)-β-alanine (DGTA).

DGTS (in combination with DGLA) from *Lobosphaera incisa* was reported to inhibit the NF-kB-mediated pathway. This effect was attributed to an interference with the upstream inhibitor of κB (IkB) kinase complex [148], preventing the phosphorylation-dependent release of the transcriptional activator NF-kB and its nuclear translocation [149,150]. Moreover, the treatment with DGTS extracted from *Nannochloropsis granulate* caused a strong downregulation of the expression of the pro-inflammatory gene inducible nitric oxide synthase (*Nos2*) in cultured activated macrophages [151,152]. Similarly, the chlorodendrophyte *Tetraselmis chui* was recently reported to produce monogalactosyldiacylglycerols with a strong inhibitory effect on nitric oxide synthesis [153].

### Enhancement of DHA and EPA Through Culture Optimization and Genetic Engineering

At present, a major limitation to the widespread use of marine microalgal BL for human nutrition lies in the low yields and recovery of these metabolites. The industrial production of EPA and DHA from native strains can be maximized by tailoring key cultivation parameters including temperature, light intensity, and salinity, as recently reported for *Nannochloropsis oceanica*, which displayed the highest EPA accumulation at low temperature (19 °C), despite a slower growth rate and lower overall biomass accumulation [154]. Another suggested strategy to enhance cellular lipid accumulation includes CO_2_ limitation, as reported for the model green microalga *Chlamydomonas reinhardtii* (chlorophyte) [155], although it is not tested in *N. oceanica*. More recently, a co-cultivation system using the golden-brown microalga *Tisochrysis lutea* (DHA producer [156,157]) and the green microalga *Microchloropsis salina* (EPA producer) reported a higher biomass accumulation compared with monocultivation systems and the additional advantage of producing a near 1:1 nutritionally balanced ratio of DHA/EPA, the accumulation of which could be enhanced by 31 and 80% compared with monocultures, respectively [158,159].

Furthermore, the metabolic flexibility of microalgae can be exploited to establish the large-scale mixotrophic cultivation of marine strains using reduced carbon substrates to boost the accumulation of algal biomass, as reported for a number of species [160,161]. A similar approach has been reported for the chlorophyte *Tetraselmis suecica* [162], *Nannochloropsis granulata* [163], and the diatom *Phaeodactylum tricornutum* [164,165], although several other marine strains might display mixotrophic capability [166,167,168,169,170,171]. It should be noted, however, that growth enhancement does not always correlate with the over-accumulation of target metabolites [172]. Finally, the recovery of BL lipids from microalgal biomass can be further enhanced by optimizing extraction processes [173,174,175], including advanced techniques based on lipase-catalyzed hydrolysis [176].

A feasible strategy to boost the accumulation of BL precursors in phytoplankton is via the domestication of wild oleaginous microalgae. Strain improvement is routinely performed in model species by accelerating mutation rates and applying high-throughput phenotyping of lipid-related traits (directed evolution), or via adaptive laboratory evolution [177,178,179,180,181,182,183,184], although a few examples of domestication have been recently reported also for *Tisochrysis lutea* and the diatom *Nitzschia inconspicua* [185,186,187,188].

The rational genetic engineering of lipid metabolism in marine phytoplankton is also relatively advanced [189,190]. Most approaches exploit the overexpression of either heterologous lipid anabolism genes or endogenous transcription factors [191,192,193], although productivity in oleaginous microalgae can also be achieved by modulating light-use efficiency [194]. At present, *Nannochloropsis* sp. is the reference organism for the industrial production of EPA and is an established genetic model system for engineering lipid metabolism in microalgae [195,196]. The availability of a dedicated molecular toolkit enables complex genetic interventions in this oleaginous microalga, affording the stacking of transgenes and the analysis of a multiple expression cassette design via streamlined modular cloning and assembly strategies [43,197,198]. For instance, this technology was employed to genetically enhance EPA biosynthesis in the strain *N. oceanica* CCMP1779 via the overexpression of previously identified endogenous candidate fatty acid desaturase genes (*Δ12* and *Δ5 FAD*) [199], while an increase in EPA production was achieved via the overexpression of the endogenous *Δ6* isoform [200]. Two recent approaches to the genetic engineering of *N. oceanica* reported an increased valorization of the health potential of this microalga by coupling the synthesis of endogenous lipid with heterologous immunomodulatory pigments. In one case, through an iterative transgenesis strategy of β-carotenoid ketolase and hydroxylase genes, it was possible to achieve a more than 130-fold increase in the synthesis of the potent antioxidant ketocarotenoid astaxanthin [201,202,203,204], while in the second case, *N. oceanica* was engineered to over-accumulate the related antioxidant ketocarotenoid canthaxanthin [205].

Targeted genetic manipulations are now also possible in *N. oceanica* thanks to the existence of dedicated genome editing techniques [206], including a toolbox based on the Clustered Regularly Interspaced Short Palindromic Repeats (CRISPR)-Cas12 technology [207]. Notably, this system exploits an episomal vector system encoding the Cas12 nuclease and a customizable set of single guide RNAs, enabling the transient expression of the editing complex and subsequent removal, affording multiple cycles of intervention. This system has clear advantages over the delivery of unstable ribonucleoprotein complexes and avoids the stable integration of transgenes in the algal genome.

The advancement of lipid metabolism engineering requires the continuous development of novel genetic parts to build synthetic pathways, including cis-acting regulatory elements and promoters, the identification of metabolic bottlenecks though omics-based studies [208,209,210], and the transfer of technologies from model to emerging species, as recently reported for the extremophile microalga *Chlamydomonas pacifica* [211].

## 4. Prostaglandin Metabolism in Diatoms

The existence of PGs in photosynthetic organisms was firstly described in macroalgae species of the *Gracilaria* genus, in the microalga *Euglena Gracilis*, and in cyanoprokaryotes of the Oscillatoria and Microcystidaceae taxa [212,213,214,215]. The identified molecules were PGA_2_, PGB_2_, PGD_1_, PGE_1_, PGE_2_, 5-keto-PGE_2_, PGF_2α_, and PGJ_2,_ and their production was reported to correlate with light, temperature, and metal stresses [216].

The first elucidation of a complete ad active PG biosynthetic pathway in microalgae was described for two strains of the chain-forming diatom *Skeletonema marinoi* [42], and later in dinoflagellates, a vast group of microalgae with untapped biotechnological potential [42,217,218] (summarized in Figure 3).

Diatoms constitute an extremely diversified group of eukaryotic phytoplankton (clade Stramenopiles and Bacillariophyceae) of great ecophysiological relevance, with a global distribution in the oceans and accounting for an estimated 20% of annual CO_2_ fixation [219,220,221]. At present, two species, *Thalassiosira pseudonana* and *P. tricornutum*, represent the reference organisms for diatom biotechnology research, although emerging species like *Fragilariopsis cylindrus* and *Pseudo-nitzschia multistriata* are being explored as model organisms for basic and applied research. Remarkably, several diatom species undergo dramatic reprogramming of their lipid metabolism upon environmental stresses, mainly nutrient starvation [222,223,224,225,226], and have recently emerged as natural producers of EPA and DHA [227,228,229,230,231].

At present, the two centric diatoms *S. marinoi* and *Thalassiosira rotula* are used as model organisms for PG biosynthesis, although the information contained in the Marine Microbial Eukaryote Sequencing Project database (MOORE) [232] suggests the expression of PG biosynthetic genes in several other species and phylogenetic groups [42]. For instance, the mining of dinoflagellate transcriptomes identified genes with a putative involvement in both PG biosynthesis and catabolism, including the PG terminal synthases for PGE and PGD (PTGES and PTGDS) and the catabolic enzyme PG reductase 1, although the limiting enzyme COX-1 was not found [217].

Studies on *S. marinoi* and *T. rotula* revealed the existence of a canonical animal PG pathway in unicellular eukaryotes and their highly regulated biosynthesis, possibly as a defense response to grazers [40,42,233]. Both species expressed the COX-1 gene during the transition between the exponential and stationary growth phases and downregulated its transcription in the senescent phase, while the levels of two terminal synthases PTGES and PTGDS did not vary. Instead, PTGES and PTGDS displayed a differential regulation between two *S. marinoi* strains [42]. While strain FE7, a high oxylipin producer, showed a strong expression of all PG enzymes and downregulated COX-1 during senescence, strain FE60, a low oxylipins producer, downregulated both PTGES and PTGDS in the stationary phase and displayed an overall lower expression of all PG genes. Those differences were also reflected by the cellular content of the main EPA metabolite, PGE_3_, which resulted six times higher in strain FE7 in the senescent phase compared with strain FE60, as also confirmed in a more recent analysis [234].

Furthermore, it was shown that the expression of the COX-1, PTGES, and the PTGFS genes was downregulated in *T. rotula* in response to nutrient starvation, particularly of silica [39], with a stronger effect observed in a strain native to the Mediterranean Sea compared with other isolates.

The discovery of PG metabolism in diatoms suggests the possibility to use these organisms for the biotechnological production of these BLs as alternative strategy to the current approaches. Indeed, the industrial PG production is not straightforward [235]. For instance, chemical synthesis is complicated by the instability of intermediate and final molecules, and requires several chemical reactions [236]. The heterologous production of PGs, instead, has been reported in the red macroalgae *Gracilaria vermiculophylla* [237] and *Agarophyton vermiculophyllum* [238], in the model angiosperm *Arabidopsis thaliana* [239], in the liverwort *Marchantia polymorpha* [240], and in diatom species lacking COX genes [17], although the yields did not satisfy cost-effectiveness.

### Genetic Engineering of Bioactive Lipid Production in Diatoms

Genetic engineering in diatoms is established and several works have already investigated lipid metabolism in *P. tricornutum* by means of gene silencing and targeted gene inactivation of endogenous lipases [241,242,243,244]. Homology-directed gene replacement could also be established in *P. tricornutum* bypassing the non-homologous end joining (NHEJ) DNA repair pathway responsible for foreign DNA integration at random positions in the diploid karyome. This was possible by knockdown of the endogenous DNA-Ligase IV (LigIV) gene, resulting in a more efficient homologous recombination at the triacylglycerol lipase 1 (tgl1) locus [245].

Gene-editing interventions have been reported in diatoms using transcription activator-like effector nucleases (TALEN) [246], the CRISPR method using a Cas9 nuclease coupled with transgene transfection [247], and even more advanced DNA-free strategies using ribonucleoprotein complexes [248,249]. More recently, a transient system based on self-replicating episomes delivered via bacterial conjugation was implemented in *P. tricornutum*, affording the simultaneous expression of the Cas9 nuclease and of metabolic enzymes [250]. Moreover, efficient gene editing was performed in the emerging model species *T. pseudonana* using a plasmid-encoded Cas9 nucleases [251] and CRISPR/Cas-mediated homologous recombination [221,252].

Multiplex gene editing was also reported in *P. tricornutum* to alter the cellular lipid droplet content. This strategy exploited a self-replicating episome encoding a Cas9 nuclease and a single guide (sgRNA) RNA array targeting the stramenopile-type lipid droplet protein (StLDP; Phatr3_J48859), a key factor regulating droplet size and lipid homeostasis [253]. This approach produced high-frequency biallelic *StLDP* knock-out mutants with oversized lipid droplets upon nitrogen depletion, which were not degraded upon nutrient repletion, therefore paving the way for future rational genetic interventions to enhance the cellular content of omega-3 fatty acids in diatoms. Furthermore, a protein termed acyl-CoA binding protein (PtACBP) was identified *P. tricornutum*. This factor appears to regulate lipid droplet degradation and the biosynthesis of EPA [254], thus representing a novel target for genetically modulated diatom lipid metabolism. Another recently identified target for enhancing DHA biosynthesis in diatoms is the Δ5 elongase gene (*ptELO5a*), whose inactivation via CRISPR-Cas9-based genome editing significantly reduced the intracellular levels of this BL precursor [255]. In this respect, a synthetic biology approach was recently reported in *P. tricornutum* [165] via heterologous over-expression of the Δ5-elongase (*Ppglut1*) gene from the chlorophyte *Ostreococcus tauri* and of a glucose transporter (OtElo5) from the moss *Physcomitrella patens* to enable mixotrophic cultivation and the enhanced production of DHA. Indeed, precursor limitation was reported in a previous study on *P. tricornutum*, in which the over-expression of the endogenous diacylglycerol acyltransferase 2 (DGAT2) gene, a key enzyme in triacylglycerol biosynthesis, enhanced cellular neutral lipid content and resulted in almost a 80% increase in EPA production [192].

Although the molecular toolkits and gene editing techniques for nuclear gene inactivation and transgenesis in microalgae are growing [256,257,258], the repertoire of genetic parts, including promoters and other cis-acting regulatory elements, and selectable markers in diatoms is still scarce. Also, organellar genetic engineering in haptophytes is still lagging far behind that in their freshwater, green counterparts [259,260,261,262]. Improvements in this area are, therefore, needed and expected to foster the biotechnological exploitation of diatoms [263]. Moreover, it is of paramount importance to develop genetic toolboxes for emerging species with biotechnological potential, as reported for *S. marinoi* [264].

## 5. Conclusions and Outlook

The biotechnological potential of marine microalgae for producing BLs is fully established. The safety profile and molecular understanding of the mechanisms of action of microalgal BLs supports continuous innovation in this research field to develop streamlined production platforms of functional lipids with clinical applications in IMIDs and human nutrition. To further expand the clinical uses of microalgal lipids, it is, however, necessary to expand the repertoire of useful molecules and test their effectiveness in vitro, in vivo, and finally, in human clinical trials.

Therefore, a detailed characterization of the main classes of non-enzymatic oxylipins produced in microalgae is urged to further expand the repertoire of microalgal BLs and enable their use in clinical research. To this end, a continuous bioprospecting activity is essential to identify novel native producers and test their cultivation under different conditions to assess how BL yields are affected.

Lastly, rigorous practices and regulatory frameworks should be implemented in the industrial production and downstream processing of microalgal biomass, including from genetically engineered strains, to ensure its biosafety and, thus, promote the market acceptance of microalgae-derived functional foods by consumers and health specialists.

## Figures and Tables

**Figure 1 marinedrugs-23-00086-f001:**
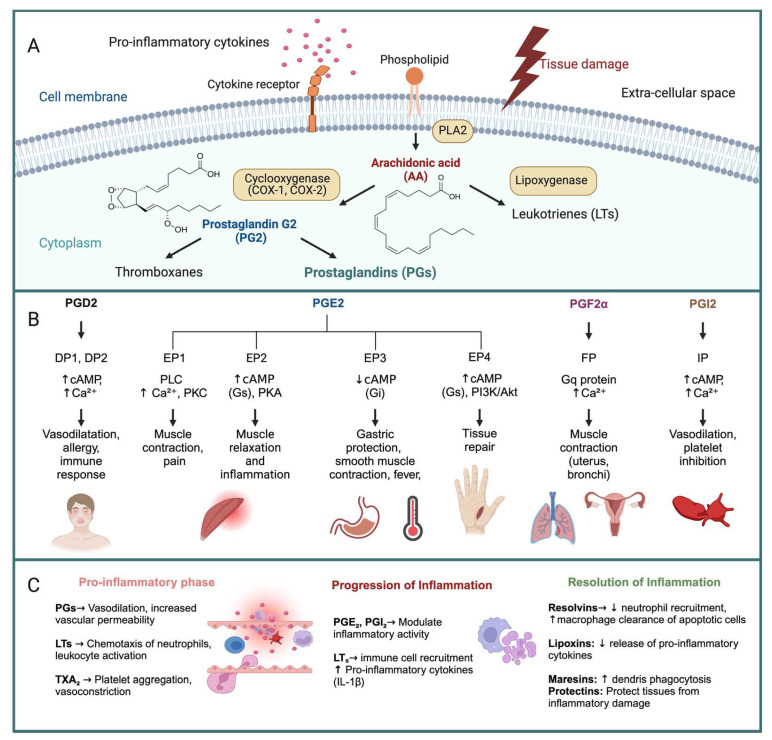
Biosynthesis and roles of bioactive lipids in human physiology. (**A**) Upon tissue damage and signaling derived from pro-inflammatory cytokines binding to membrane receptors, membrane phosholipids are substrates of phospholipase enzymes (PLA2) that release arachidonic acid (AA). AA is further converted into prostaglandins (PGs) by cyclooxygenase enzymes (COX1 and COX2) and into leukotrienes (LTs) by lipooxygenase enzymes. PGs such as PG2 can be further converted into thromboxanes. BLs are secreted and regulate the initiation, progression, and resolution of inflammatory responses (described in (**C**)). (**B**) PG molecules exert different functions depending on which receptor type they bind, leading to multiple physiological outcomes. (**C**) Inflammatory responses typically develop via a pro-inflammatory phase then progress to an acute phase and are eventually resolved. Abbreviations: PGs: prostaglandins; PLA2: phospholipase A2; COX-1: cyclooxygenase 2; COX-2: cyclooxygenase 2; PG2: prostaglandin G2; PGD2: prostaglandin D2; PGE2: prostaglandin E2; PGF2α: prostaglandin F2 alpha; PGI2: prostaglandin I2 (prostacyclin); DP1/DP2: D-prostanoid receptor 1/2; EP1/EP2/EP3/EP4: E-prostanoid receptor 1/2/3/4; FP: F-prostanoid receptor; IP: I-prostanoid receptor; PLC: phospholipase C; PKC: protein kinase C; PKA: protein kinase A; PI3K/Akt: phosphoinositide 3-kinase/protein kinase B; cAMP: cyclic adenosine monophosphate; Gs: stimulatory G protein; Gi: inhibitory G protein.

**Figure 2 marinedrugs-23-00086-f002:**
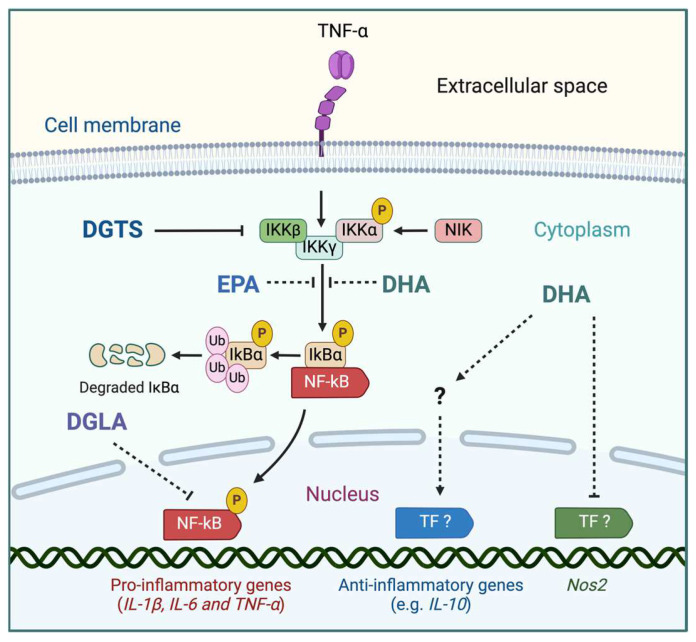
Described anti-inflammatory effects of microalgal bioactive lipids on intracellular signaling pathways. Reproduced from [34]. Pro-inflammatory cytokines such as the tumor necrosis factor α (TNF-α) activate the nuclear factor kappa Β (NF-kB)-mediated pathway, which positively regulates the expression of pro-inflammatory genes. Several microalgal lipids, including eicosapentaenoic acid (EPA), docosahexaenoic acid (DHA), and dihomo γ-linolenic acid (20:3, *n* − 6, DGLA), have been shown to interfere with intermediate steps of the signaling pathways, suppressing pro-inflammatory responses. EPA, DHA, and DGLA all appear to negatively modulate the NF-kB pathway by inhibiting the upstream inhibitor of κB (IkB) kinase complex. These compounds appear to prevent the phosphorylation-dependent release of the pro-inflammatory transcriptional activator NF-kB and its nuclear translocation, thereby suppressing the expression of pro-inflammatory genes, including the inducible nitric oxide synthase (*Nos2*) gene. DHA appears to also positively regulate the expression of the anti-inflammatory gene *IL-10*. Solid blunt arrows indicate experimentally described interference mechanisms; dashed blunt arrows indicate suggested mechanisms of action.

**Figure 3 marinedrugs-23-00086-f003:**
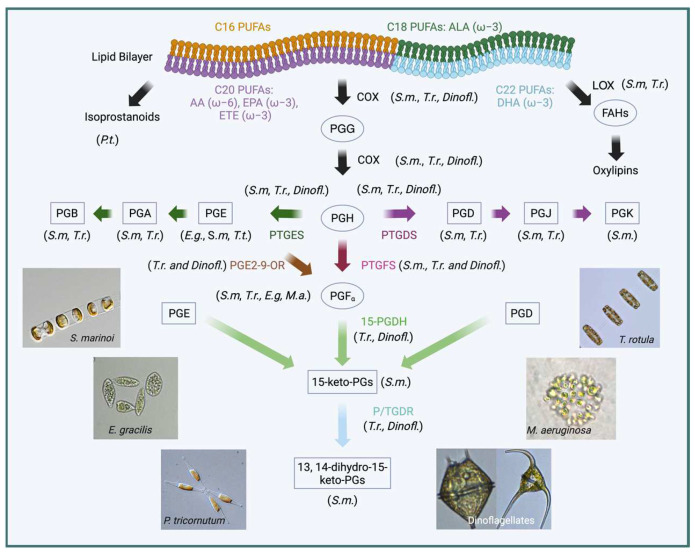
Prostaglandin metabolism in microalgae. Prostaglandins (PGs) have been described in the following microalgal species: *Skeletonema marinoi* (*S.m.*), in which the genes encoding enzymes involved in the PG pathway were identified, along with the chemical characterization of PUFA-derivative molecules; *Thalassiosira rotula* (*T.r.*), in which additional PG-related genes were identified together with PGs released in the cultivation medium; *Phaeodactylum tricornutum* (*P.t.*), in which isoprostanoids produced via isomerization of PUFA precursors were characterized; dinoflagellates species (*Dinofl.*), from which the transcriptomics data mining revealed the existence of the PG prostaglandin pathway; and the green microalga *Euglena graclilis* (*E.g.*) and the cyanobacterium *Microcystis aeruginosa* (*M.a*.), which also produce PGs. The species in which PG biosynthetic enzymes and products have been identified are indicated to the side of each figure element. Prostaglandin B (PGB); Prostaglandin A (PGA); Prostaglandin E (PGE); Prostaglandin D (PGD); Prostaglandin J (PGJ); Prostaglandin K (PJK); prostaglandin F_α_ (PGF_α_); 15-keto prostaglandin (15-keto-PGs); 13-dihydro-15-keto prostaglandin (13-dihydro-15-keto-PGs); 14-dihydro-15-keto prostaglandin (14-dihydro-15-keto-PGs); lipoxygenase (LOX); cyclooxygenase (COX); prostaglandin E synthase (PTGES); prostaglandin D synthase (PTGDS); prostaglandin F synthase (PTGFS); prostaglandin E2-9-oxoreductase (PGE2-9-OR, in the case of *T. r.*); 15-prostaglandin dehydrogenase (15-PGDH); prostaglandin reductase (PTGR). In the cases of *T. r*., PGFα and PGJ have not been directly identified, and only their degradation products have. α-Linolenic acid, ALA; eicosatrienoic acid, ETE; eicosapentaenoic acid, EPA; docosahexaenoic acid, DHA.

## Data Availability

No new data were created.

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
