# Peer review of "Marine Phytoplankton Bioactive Lipids and Their Perspectives in Clinical Inflammation"

_marinedrugs, 2025, doi:10.3390/md23020086_

Round 1
Reviewer 1 Report
Comments and Suggestions for Authors
General comments:
The paper entitled "Marine Microalgal Bioactive Lipids and Their Perspectives in Clinical Inflammation” reviews recent work on immunomodulatory lipid mediators derived from marine microalgae and their potential for the management of inflammation related disorders. Although the paper provides an interesting overview of the prostaglandin metabolism in diatoms, it hardly describes the potential of these mediators in modulating inflammation parameters, how those parameters relate to the different phases of inflammation and how that prevents the development of inflammatory related diseases. Furthermore, this paper refers to marine microalgae but focuses its overview only on diatoms failing to address the diversity in marine microalgae. Also the authors define oxylipins as bioactive lipids, which is not a problem, but then only address enzymatic oxylipins not addressing the diversity of oxygenated fatty acids metabolites, some of which have been described with promising bioactivities. Other critical points that I believe should be improved are the genetic engineering sub-chapter that focuses too much on TAG synthesis improvement and does not provide any studies on enhancement of EPA and DHA production, precursors to these bioactive lipids – also not describing how enhancement of these two precursors enhances the production of oxylipins.
This review paper holds enough potential for publication however I believe it requires some major revision, specifically some improvement in the content of each chapter to better address the topics it proposes to overview. Aside from my general comments I have some suggestions and smaller comments detailed below.
Specific comments:
Please use italics when referring to microalgae species and genus.
Line 64 – lysoglycerophospholipids and sphingolipids are not eicosanoids please clarify in the text.
Lines 65-66 – LC-PUFAs often refers to long-chain polyunsaturated fatty acids and not just omega-3 please provide a different abbreviation.
Lines 67 -68 This idea is very vague please develop on what healthier lipid profiles are and what conditions does it lower incidence/risk.
Lines 80-82 the definition of prostaglandins is very vague. I would explore more the role of prostaglandins in humans and the idea of the diversity of animal-like prostaglandins observed in microalgae, not just in diatoms.
Line 97 – Eicosanoids are oxylipins but not all oxylipins are eicosanoids.
Line 144 – Please check for the lack of space between words and full stops and double spacing.
Line 130 – Correct “cel” to cell.
Line 120-132 – The authors should develop the topic of the dynamic of different prostaglandins and interchange of COX 1/COX-2 activity during the different phases of inflammation.
Line 138 – Define PGF as I believe it wasn’t previously defined.
Lines 151-152 – The authors wrote resolvins twice. Did they mean protectins the second time?
Line 174 – I would recommend the authors to review the entire paper and use italics appropriately for microalgae species and genus.
Line 205 – Correct “ssp” to sp.
Line 206 – Nannochlorpsis species are known to produce high amounts of EPA not DHA. Why would this microalga be an organism of choice for the industrial production of DHA? Why not another microalga that produces higher amounts of DHA like Tisochrysis sp.?
Lines 238-239 – Enhancing growth of microalgae at industrial scale does not necessarily mean enhancing the production of target metabolites. Please clarify if this was addressed in this study and if the authors showed data of an enhanced production of bioactive lipids when using glycerol as reduced carbon substrate.
Lines 285-287 and lines 297-299 - Please refrain from using non-published references as data is not included.
Line 291-292 – Please provide a reference to this paragraph.
Line 299 – Please use italics for in vivo and in vitro.
Line 300 – I believe it would make more sense to join this paragraph with the previous three from lines 285-288, lines 289-292 and lines 293-296.
Lines 307-315 – I would also join these two paragraphs.
Lines 316 - This chapter is entitled “Enhancement of DHA and EPA through genetic engineering” but more than half of it refers to TAG production and not EPA or DHA. DHA enhancement is only mentioned in line 361. Also, it is not explained how the enhancement of DHA production would enhance in turn the production of bioactive lipids (oxylipins) derived from DHA. Particularly, enzymatically derived oxylipins. I believe the authors should provide other studies backing enhancement of EPA and DHA production through genetic engineering or remove this entire chapter.
Lines 319 “and named” remove italics. Please make sure that italics are properly used throughout the text – mentioned of microalgae, in vivo and in vitro and to write genes as “tgl1”
I believe this paper would benefit from a wider overview. Oxylipins can be obtained enzymatically and non-enzymatically, and the oxylipins referred by the authors in this review are enzymatic oxylipins. Non-enzymatic oxylipins from microalgae have also been described with interesting bioactive properties and could bring fort novel ingredients/compounds for clinical applications. I think this review would benefit from summarizing recent works on these bioactive lipids.

Comments on the Quality of English Language
I would recommend the authors to improve the English throughout the text.
Author Response
General comments:
The paper entitled "Marine Microalgal Bioactive Lipids and Their Perspectives in Clinical Inflammation” reviews recent work on immunomodulatory lipid mediators derived from marine microalgae and their potential for the management of inflammation related disorders. Although the paper provides an interesting overview of the prostaglandin metabolism in diatoms, it hardly describes the potential of these mediators in modulating inflammation parameters, how those parameters relate to the different phases of inflammation and how that prevents the development of inflammatory related diseases.
EAC: thank you for this observation. We have now implemented a discussion on the physiological processes in which PGs are involved. Now lines 138-153
Furthermore, this paper refers to marine microalgae but focuses its overview only on diatoms failing to address the diversity in marine microalgae. Also the authors define oxylipins as bioactive lipids, which is not a problem, but then only address enzymatic oxylipins not addressing the diversity of oxygenated fatty acids metabolites, some of which have been described with promising bioactivities.
EAC: We have added a short discussion on the occurrence of non-enzymatically produced oxylipins in microalgae now lines: 382-386
We also described betaine lipids found in microalgal species and their potential use in the clinical practice now lines: 198-207/214-218
Other critical points that I believe should be improved are the genetic engineering sub-chapter that focuses too much on TAG synthesis improvement and does not provide any studies on enhancement of EPA and DHA production, precursors to these bioactive lipids – also not describing how enhancement of these two precursors enhances the production of oxylipins.
EAC: we have now included a short discussion on the reported enhancement of EPA and DHA via genetic engineering. Now lines 272-276/436-443
This review paper holds enough potential for publication however I believe it requires some major revision, specifically some improvement in the content of each chapter to better address the topics it proposes to overview. Aside from my general comments I have some suggestions and smaller comments detailed below.
EAC: Thank you for acknowledging the value of our review paper and for your time in revising our manuscript and providing useful inputs to improve it.
We have also included three figures in which we described the roles of bioactive lipids in human physiology and disease, the intracellular effects of immunomodulatory lipids from microalgae and the biosynthesis of PGs in diatoms species.
Specific comments:
Please use italics when referring to microalgae species and genus.
EAC: We have corrected that throughout the manuscript
Line 64 – lysoglycerophospholipids and sphingolipids are not eicosanoids please clarify in the text.
EAC: we have now amended this definition
Lines 65-66 – LC-PUFAs often refers to long-chain polyunsaturated fatty acids and not just omega-3 please provide a different abbreviation.
EAC: we have rephrased this sentence accordingly
Lines 67 -68 This idea is very vague please develop on what healthier lipid profiles are and what conditions does it lower incidence/risk.
EAC: we have rephrased this sentence and included appropriate references lines
Lines 80-82 the definition of prostaglandins is very vague. I would explore more the role of prostaglandins in humans and the idea of the diversity of animal-like prostaglandins observed in microalgae, not just in diatoms.
EAC: We have now implemented a discussion on the physiological processes in which PGs are involved. Now lines 138-153
Line 97 – Eicosanoids are oxylipins but not all oxylipins are eicosanoids.
EAC: We have removed oxylipins from this line and used this term correctly elsewhere in the text
Line 144 – Please check for the lack of space between words and full stops and double spacing.Line
EAC: done
130 – Correct “cel” to cell.
EAC: done
Line 120-132 – The authors should develop the topic of the dynamic of different prostaglandins and interchange of COX 1/COX-2 activity during the different phases of inflammation.
EAC: We have expanded the discussion on the physiological processes in which different PGs are involved depending on which receptor type they bind. We also improved the discussion of the differential roles of COX1/2 enzymes. Now lines 139-153
Line 138 – Define PGF as I believe it wasn’t previously defined.
EAC: done
Lines 151-152 – The authors wrote resolvins twice. Did they mean protectins the second time?
EAC: corrected
Line 174 – I would recommend the authors to review the entire paper and use italics appropriately for microalgae species and genus.
EAC: done
Line 205 – Correct “ssp” to sp.
EAC: done
Line 206 – Nannochlorpsis species are known to produce high amounts of EPA not DHA. Why would this microalga be an organism of choice for the industrial production of DHA? Why not another microalga that produces higher amounts of DHA like sp.?
EAC: we have rectified this statement and added appropriate references new lines: 239-242/265-268
Lines 238-239 – Enhancing growth of microalgae at industrial scale does not necessarily mean enhancing the production of target metabolites. Please clarify if this was addressed in this study and if the authors showed data of an enhanced production of bioactive lipids when using glycerol as reduced carbon substrate.
EAC: we have moved this reference to the conclusions section together with others and specified that it should be checked if the lipid profiles vary under mixotrophic growth conditions. New lines: 463-469
Lines 285-287 and lines 297-299 - Please refrain from using non-published references as data is not included.
EAC: we have removed these statements throughout the text.
Line 291-292 – Please provide a reference to this paragraph.
EAC: A reference was added to this paragraph.
Line 299 – Please use italics for in vivo and in vitro.
EAC: done
Line 300 – I believe it would make more sense to join this paragraph with the previous three from lines 285-288, lines 289-292 and lines 293-296.
EAC: thank you for this suggestion, we have joined these parts
lines:
Lines 307-315 – I would also join these two paragraphs.
EAC: thank you for this suggestion, we have joined these parts
lines:
Lines 316 - This chapter is entitled “Enhancement of DHA and EPA through genetic engineering” but more than half of it refers to TAG production and not EPA or DHA. DHA enhancement is only mentioned in line 361.
EAC: thank you for this observation. However, there was an editing mistake from our side since chapter 4.1 and 3.1 were titled same. Paragraph 4.1 now is titled “Genetic Engineering of Lipid Metabolism in Diatoms, which truly reflects the content of this section.
Also, it is not explained how the enhancement of DHA production would enhance in turn the production of bioactive lipids (oxylipins) derived from DHA. Particularly, enzymatically derived oxylipins. I believe the authors should provide other studies backing enhancement of EPA and DHA production through genetic engineering or remove this entire chapter.
EAC: We have added a short discussion on the occurrence of non-enzymatically produced oxylipins in microalgae now lines: 463-469
Lines 319 “and named” remove italics. Please make sure that italics are properly used throughout the text – mentioned of microalgae, in vivo and in vitro and to write genes as “tgl1”
EAC: done
I believe this paper would benefit from a wider overview. Oxylipins can be obtained enzymatically and non-enzymatically, and the oxylipins referred by the authors in this review are enzymatic oxylipins. Non-enzymatic oxylipins from microalgae have also been described with interesting bioactive properties and could bring fort novel ingredients/compounds for clinical applications. I think this review would benefit from summarizing recent works on these bioactive lipids.
EAC: We have added a short discussion on the occurrence of non-enzymatically produced oxylipins in microalgae now lines: 463-469
Reviewer 2 Report
Comments and Suggestions for Authors
Reviewer’s comments concerning the manuscript entitled: “Marine Microalgal Bioactive Lipids and Their Perspectives in Clinical Inflammation”
This is a review article in which the authors wanted to present the latest knowledge on bioactive lipids found in marine microalgae and their potential role in clinical inflammation. The subject appears to be novel and of interest. It is related to the scope of “Marine Drugs”. Generally, the manuscript is well-written and worth publishing in this journal, but with a few necessary revisions. First of all, in its current form, the article does not fully reflect the proposed title. The authors should have elaborated on the section on clinical inflammation, while they focused on genetic modifications to increase the synthesis of docosahexaenoic and eicosapentaenoic acids (DHA and EPA, respectivelly). Genetic engineering is a controversial subject and so far, oil extracted from marine microalgae is only used as biofuel. In this light, the subsection “Enhancement of DHA and EPA Through Genetic Engineering” does not match the chapter “Prostaglandin Metabolism in Diatoms”. The authors should discus the eventual use of DHA and EPA obtained through genetic engineering of microalgae genes in terms of benefits and risks to human health.
Below there are some other issues in the manuscript, that have to be addressed and corrected:
Lines 62-64: “The main classes of BL involved in the initiation of inflammation and progression towards chronic conditions are the eicosanoids, among which are the resolvins (specialized pro-resolving mediators), prostaglandins, […]”.
From this sentence, the reader may get the impression that resolvins are involved in the development of inflammation, whereas they are strong anti-inflammatory compounds.
Lines 90-95: “The introduction should briefly place the study in a broad context and highlight why it is important. It should define the purpose of the work and its significance. The current state of the research field should be carefully reviewed and key publications cited. Please highlight controversial and […]”.
This is the Introduction part from the template file that the authors should have removed. However, I totally agree with these indications.
Lines 97-98: “Eicosanoids (or oxylipins) constitute the main class of signaling BL involved in the initiation and resolution of inflammation”.
This is a very general statement, because the same eicosanoids are not involved both in initiation and resolution of inflammation.
Lines 101-103: Eicosanoids comprise different classes of molecules, among which prostaglandins (PGs), prostacyclins, thromboxanes (TXs), and leukotrienes (LTs) are collectively known as prostanoids”.
Prostanoids are metabolites of 20-carbon fatty acids formed on the path of cyclooxygenases. Meanwhile, leukotrienes are formed via the lipoxygenase pathway and thus do not belong to the prostanoids.
Lines 112-116: “PGs derive from AA and regulate multiple physiological processes, from blood flow regulation to inflammation and immune responses, but also to cause disease states [27].Collectively, the AA cascade is central to the inflammatory process, as it produces, besides PGs, TXs which promote platelet aggregation and vasoconstriction, and LTs, which enhance vascular permeability [28]”.
Eicosanoids also derived from eicosapentaenoic acid (EPA) and dihomo-γ-linolenic acid (DGLA) and this should be mentioned. The authors should give some examples of PGs synthesized from AA, EPA and DGLA. What are the roles of PGs derivatized from EPA and DGLA?
The manuscript should be carefully checked for spelling and punctuation, as minor errors such as double spaces (e.g. line 77) and missing punctuation marks (e.g. line 75) or words (e.g. line 106 – missing “in” in the expression “PGs were discovered the 1930s”) occur. Latin names of microalgae as well as expressions such as “in vitro” (e.g. line 176) and “in vivo” should be italicized.
Author Response
This is a review article in which the authors wanted to present the latest knowledge on bioactive lipids found in marine microalgae and their potential role in clinical inflammation. The subject appears to be novel and of interest. It is related to the scope of “Marine Drugs”. Generally, the manuscript is well-written and worth publishing in this journal, but with a few necessary revisions. First of all, in its current form, the article does not fully reflect the proposed title. The authors should have elaborated on the section on clinical inflammation, while they focused on genetic modifications to increase the synthesis of docosahexaenoic and eicosapentaenoic acids (DHA and EPA, respectivelly).
EAC: Thank you for acknowledging the value of our review paper and for your time in revising our manuscript and providing useful inputs to improve it. In the revised version of the manuscript, we have elaborated the part on the clinical aspect of bioactive lipids and expanded the preclinical evidence on the effects of microalgae-derived lipids on the modulation of proinflammatory responses via interference on the intracellular cascades. Lines: 90-114
We have also included three figures in which we described the roles of bioactive lipids in human physiology and disease, the intracellular effects of immunomodulatory lipids from microalgae and the biosynthesis of PGs in diatoms species.
Genetic engineering is a controversial subject and so far, oil extracted from marine microalgae is only used as biofuel. In this light, the subsection “Enhancement of DHA and EPA Through Genetic Engineering” does not match the chapter “Prostaglandin Metabolism in Diatoms”. The authors should discus the eventual use of DHA and EPA obtained through genetic engineering of microalgae genes in terms of benefits and risks to human health.
EAC: Thank you for this observation. However, there was an editing mistake from our side since chapter 4.1 and 3.1 were titled same. Paragraph 4.1 now is titled “Genetic Engineering of Lipid Metabolism in Diatoms, which truly reflects the content of this section.Despite the controversial use of genetic engineering for food-feed related products, we believe it is worth acknowledging the efforts that are being made in the strain improvement and the development of state-of-the art genetic tools, including DNA-free gene editing approaches. We have now stated in the conclusions that regulatory frameworks should be put in force to guarantee safety of food products derived from genetically engineered microalgae.
Below there are some other issues in the manuscript, that have to be addressed and corrected:
Lines 62-64: “The main classes of BL involved in the initiation of inflammation and progression towards chronic conditions are the eicosanoids, among which are the resolvins (specialized pro-resolving mediators), prostaglandins, […]”.
From this sentence, the reader may get the impression that resolvins are involved in the development of inflammation, whereas they are strong anti-inflammatory compounds.
EAC: Thank you for pointing out that. We have amended this sentence.
Lines 90-95: “The introduction should briefly place the study in a broad context and highlight why it is important. It should define the purpose of the work and its significance. The current state of the research field should be carefully reviewed and key publications cited. Please highlight controversial and […]”.
This is the Introduction part from the template file that the authors should have removed. However, I totally agree with these indications.
EAC: Done
Lines 97-98: “Eicosanoids (or oxylipins) constitute the main class of signaling BL involved in the initiation and resolution of inflammation”.
This is a very general statement, because the same eicosanoids are not involved both in initiation and resolution of inflammation.
EAC: We have amended this sentence and provided a better description. Lines: 139-153
Lines 101-103: Eicosanoids comprise different classes of molecules, among which prostaglandins (PGs), prostacyclins, thromboxanes (TXs), and leukotrienes (LTs) are collectively known as prostanoids”.
Prostanoids are metabolites of 20-carbon fatty acids formed on the path of cyclooxygenases. Meanwhile, leukotrienes are formed via the lipoxygenase pathway and thus do not belong to the prostanoids.
EAC: We have amended this sentence and provided a better description. Lines: 90-114
Lines 112-116: “PGs derive from AA and regulate multiple physiological processes, from blood flow regulation to inflammation and immune responses, but also to cause disease states [27].Collectively, the AA cascade is central to the inflammatory process, as it produces, besides PGs, TXs which promote platelet aggregation and vasoconstriction, and LTs, which enhance vascular permeability [28]”.
Eicosanoids also derived from eicosapentaenoic acid (EPA) and dihomo-γ-linolenic acid (DGLA) and this should be mentioned. The authors should give some examples of PGs synthesized from AA, EPA and DGLA. What are the roles of PGs derivatized from EPA and DGLA?
EAC: we have included the information about eicosanoid biosynthesis from EPA and DGLA and expanded the description of the physiological roles of prostaglandins, including examples of preclinical studies investigating the effects of DGLA and DGTS derived form microalgae. Lines: 90-114
The manuscript should be carefully checked for spelling and punctuation, as minor errors such as double spaces (e.g. line 77) and missing punctuation marks (e.g. line 75) or words (e.g. line 106 – missing “in” in the expression “PGs were discovered the 1930s”) occur. Latin names of microalgae as well as expressions such as “in vitro” (e.g. line 176) and “in vivo” should be italicized.
EAC: done
Reviewer 3 Report
Comments and Suggestions for Authors
This is a fairly laconic, but interesting review of an important and promising topic about the use of bioactive lipids from microalgae for the treatment of chronic human diseases associated with disturbances in the course of inflammatory reactions. The text is easy to read. Although, given the vastness of the available data, the text fits into 8 typewritten sheets, the authors have worked through a significant amount of literature and covered the major points and topics.
The manuscript has only one major point that must be eliminated. It does not contain any illustrative material. Meanwhile, rather complex schemes of biosynthesis of bioactive molecules and their precursors, as well as genetic engineering manipulations, require clarity. The reviewer strongly recommends that the authors make several illustrations (colorful and clear schemes) for the main parts of the manuscript.
Minor points:
Lanes 90-95 - The authors of the manuscript accidentally did not notice and did not remove the instructions on how to correctly write the Introduction. Please delete them.
Lanes 161-163 – The same remark. Please delete the instructions.
Please also check the text carefully for typos such as Euglena Gracilis - Lane 246.
Author Response
This is a fairly laconic, but interesting review of an important and promising topic about the use of bioactive lipids from microalgae for the treatment of chronic human diseases associated with disturbances in the course of inflammatory reactions. The text is easy to read. Although, given the vastness of the available data, the text fits into 8 typewritten sheets, the authors have worked through a significant amount of literature and covered the major points and topics.
The manuscript has only one major point that must be eliminated. It does not contain any illustrative material. Meanwhile, rather complex schemes of biosynthesis of bioactive molecules and their precursors, as well as genetic engineering manipulations, require clarity. The reviewer strongly recommends that the authors make several illustrations (colorful and clear schemes) for the main parts of the manuscript.
EAC: Thank you for acknowledging the value of our review paper and for your time in revising our manuscript and providing useful inputs to improve it. In the newly resubmitted version, we have included three figures in which we described the roles of bioactive lipids in human physiology and disease, the intracellular effects of immunomodulatory lipids from microalgae and the biosynthesis of PGs in diatoms species.
Minor points:
Lanes 90-95 - The authors of the manuscript accidentally did not notice and did not remove the instructions on how to correctly write the Introduction. Please delete them.
EAC: thank you for this observation. This was removed.
Lanes 161-163 – The same remark. Please delete the instructions.
EAC: thank you for this observation. This was removed.
Please also check the text carefully for typos such as Euglena Gracilis - Lane 246.
EAC: thank you for this observation. This was corrected.
Round 2
Reviewer 1 Report
Comments and Suggestions for Authors
The authors have improved the manuscript according to the provided revisions. However, I still have some questions, suggestions and check points I would like the authors to address.
General comments:
- Please revise the manuscript for:
- English corrections
- Correct and incorrect use of italics
- How the authors mention genes. Usually genes are described as follows: iNOS (inducible nitric oxide synthase) gene is Nos2 with first letter in uppercase, remaining letter lowercase and in italics.
- Abbreviations introduction – please check if you have defined each abbreviation only once and at the first use of said abbreviation.
- Plural and singular words – microalgae is plural while microalga is singular – please recheck the text.
- I don’t understand the context of the last chapter before the conclusions. This chapter focuses too much on genetic engineering of TAG production, however, TAG are not described as bioactive lipids, nor are they mentioned in the text as reservoirs of BLs (like EPA). Also an improvement in TAG production does not necessarily mean an improvement in EPA production. As the authors do not specify this it is unclear which TAG species are increasing under these genetically modified microalgae.
Specific comments:
Line 76 – do the authors mean produce? The use of derive here is not appropriate.
Line 79 – “platforms” and not “platform”
Line 91 – the expression “exotic lipids” does not sound appropriate, I would change to less common lipids.
Line 98 – when representing omega like dihomo gamma-linolenic acid (20:3 n-6) the n is in italics.
Line 130 – tissue damage?
Line 137 – “bind to different cell types” and not “bind to on different”
Line 151 – Needs a full stop after “blood flow”.
Lines 157-158 – Provide examples for one or two other receptors.
Lines 158-160 - Rewrite because I believe the meaning of this sentence does not match what the authors are trying to describe. The authors were describing different biological effects of PGE2 upon activation of different receptors. The use of "Instead" is misleading and confusing. I would write something like "Other examples of prominent prostaglandins are PGF2alpha which..."
Line 162 - I would recommend the author to join this paragraph to the previous one.
Lines 163-164 - This sentence is repetition from what is mentioned above. I would delete this.
Lines 179 – the authors mention “NSAIDs and COX-inhibitors” but aren’t NSAIDs also COX inhibitors?
Line 184 - Correct to " precursors of resolving BLs, such as EPA and DHA, could be an effective..."
Lines 191 -193 - Again, the term "Instead" is misleading and provides a different connotation and meaning to the sentence.
Line 206 – n from the omegas in italics.
Lines 206- 215 - This paragraph was little confusing. The authors changed from describing the presence of dietary lipids such as the essential FA ALA, and other important omega-3 PUFA such as EPA and DHA. However then the authors jump to betaine lipids without any context which is confusing. I think it would be interesting to mentioned some of the health benefits from these PUFA and then describe that other lipid groups are found in microalgae such as phospholipids, glycolipids and betaine lipids which are being described with anti-inflammatory and immunomodulatory properties (please check the works of Banskota such as https://doi.org/10.1080/14786419.2012.696255; https://doi.org/10.1007/s10811-012-9967-1; https://doi.org/10.1007/s10811-012-9869-2;https://doi.org/10.1080/14786419.2012.717285)
Line 237 – remove subscript for dihomo linolenic acid
Line 240 – Full stop after complex.
Line 267 - Did these authors of this study evaluate the impact on EPA and DHA? If yes then, include here.
Line 268 - Microalgae is plural as the authors are referring to one single species, the correct term should be microalga.
Line 283 – without italics.
Lines 285-292 - It is not clear wether this approach has been used in N. oceanica genes and what the results were. Please clarify.
Line 296 – without italics
Lines 314-316 - But these are prostaglandins... There is a lack of connection to this. I would suggested completing the previous idea, mention a few isoprostanoids detected in the mentioned species and then introduce the detection of prostaglandins in microalgae.
Line 340- rotula in italics
Line 347- remove italics
Line 356 – COX without italics
Lines 359 – 360 – PTGES and PTGDS without italics
Lines 361-363 - What were the PG enzymes? What was the different behavior?
Lines 364 – Which were these conserved differences?
Lines 365-368 - This has no context. Where was this analysis performed? Which microalgae? Was this a part of the previous paragraph? Was this a result of a different type of analysis? It is not clear.
Line 373 – COX, PTGES and PTGFS without italics
Line 377 – Lacks a reference.
Line 380 – COX without italics.
Lines 394-398 - Rephrase and expand in other to better understand the possibility of non-enzymatic oxylipins. Why are they more advantageous than enzymatic? Is it yields of production wise?
Line 400 - What is the relevance of editing TAG synthesis? Are TAGs bioactive lipids themselves? Are they storage of bioactive lipids? What is their relevance in the context of this review paper?
Line 425 - CRISPR has been used in page 8 and only defined here. Please recheck all the document for acronyms and abbreviations to be defined the first time they are being used.
Lines 448-450 – Please use italics appropriately – to identify the names of moss and microalgae species and not on references and beginning of sentences that do not require italics.
Comments on the Quality of English Language
The English requires improvements.
Author Response
The authors have improved the manuscript according to the provided revisions. However, I still have some questions, suggestions and check points I would like the authors to address.
General comments:
- Please revise the manuscript for:
- English corrections
- Correct and incorrect use of italics
- How the authors mention genes. Usually genes are described as follows: iNOS (inducible nitric oxide synthase) gene is Nos2 with first letter in uppercase, remaining letter lowercase and in italics.
- Abbreviations introduction – please check if you have defined each abbreviation only once and at the first use of said abbreviation.
- Plural and singular words – microalgae is plural while microalga is singular – please recheck the text.
- I don’t understand the context of the last chapter before the conclusions. This chapter focuses too much on genetic engineering of TAG production, however, TAG are not described as bioactive lipids, nor are they mentioned in the text as reservoirs of BLs (like EPA). Also an improvement in TAG production does not necessarily mean an improvement in EPA production. As the authors do not specify this it is unclear which TAG species are increasing under these genetically modified microalgae.
EAC: Thank you for your suggestions which are helpful to further improve the quality of our manuscrit. We have made correction throughout the text accordingly. We have also rearranged the chapter on genetic engineering in diatoms now titled “Genetic Engineering of Bioactive Lipid Production in Diatoms”. We shortened the part on TAG production, focussed on the available gene editing tools and only mentioned studies reporting an increase in EPA and DHA.
Specific comments:
Line 76 – do the authors mean produce? The use of derive here is not appropriate.
EAC: done
Line 79 – “platforms” and not “platform”
EAC: done
Line 91 – the expression “exotic lipids” does not sound appropriate, I would change to less common lipids.
EAC: changed to non-native lipids
Line 98 – when representing omega like dihomo gamma-linolenic acid (20:3 n-6) the n is in italics.
EAC: done
Line 130 – tissue damage?
EAC: done
Line 137 – “bind to different cell types” and not “bind to on different”
EAC: done
Line 151 – Needs a full stop after “blood flow”.
EAC: done
Lines 157-158 – Provide examples for one or two other receptors.
EAC: we have now included examples for EP2 and EP4 with appropriate references
Lines 158-160 - Rewrite because I believe the meaning of this sentence does not match what the authors are trying to describe. The authors were describing different biological effects of PGE2 upon activation of different receptors. The use of "Instead" is misleading and confusing. I would write something like "Other examples of prominent prostaglandins are PGF2alpha which..."
EAC: done
Line 162 - I would recommend the author to join this paragraph to the previous one.
EAC: done
Lines 163-164 - This sentence is repetition from what is mentioned above. I would delete this.
EAC: done
Lines 179 – the authors mention “NSAIDs and COX-inhibitors” but aren’t NSAIDs also COX inhibitors?
EAC: we have rephrased accordingly
Line 184 - Correct to " precursors of resolving BLs, such as EPA and DHA, could be an effective..."
EAC: done
Lines 191 -193 - Again, the term "Instead" is misleading and provides a different connotation and meaning to the sentence.
EAC: we have rephrased accordingly
Line 206 – n from the omegas in italics.
EAC: done
Lines 206- 215 - This paragraph was little confusing. The authors changed from describing the presence of dietary lipids such as the essential FA ALA, and other important omega-3 PUFA such as EPA and DHA. However then the authors jump to betaine lipids without any context which is confusing. I think it would be interesting to mentioned some of the health benefits from these PUFA and then describe that other lipid groups are found in microalgae such as phospholipids, glycolipids and betaine lipids which are being described with anti-inflammatory and immunomodulatory properties (please check the works of Banskota such as https://doi.org/10.1080/14786419.2012.696255; https://doi.org/10.1007/s10811-012-9967-1; https://doi.org/10.1007/s10811-012-9869-2;https://doi.org/10.1080/14786419.2012.717285)
EAC: we have accordingly rearranged this chapter by first providing experimental evidence of health benefits of EPA and DHA from microalgae and then introduced betaine lipids and their emerging pharmacological potential, using the suggested references on marine species.
Line 237 – remove subscript for dihomo linolenic acid
EAC: done
Line 240 – Full stop after complex.
EAC: done
Line 267 - Did these authors of this study evaluate the impact on EPA and DHA? If yes then, include here.
EAC: done
Line 268 - Microalgae is plural as the authors are referring to one single species, the correct term should be microalga.
EAC: done
Line 283 – without italics.
EAC: done
Lines 285-292 - It is not clear wether this approach has been used in N. oceanica genes and what the results were. Please clarify.
EAC: we have clarified this part of the text by first describing transgenesis approaches and then gene editing
Line 296 – without italics
EAC: done
Lines 314-316 - But these are prostaglandins... There is a lack of connection to this. I would suggested completing the previous idea, mention a few isoprostanoids detected in the mentioned species and then introduce the detection of prostaglandins in microalgae.
EAC: we have amended rearranged this chapter following your suggestion, including references to isoprotanoid detection in diatoms.
Line 340- rotula in italics
EAC: done
Line 347- remove italics
EAC: done
Line 356 – COX without italics
EAC: done
Lines 359 – 360 – PTGES and PTGDS without italics
EAC: done
Lines 361-363 - What were the PG enzymes? What was the different behavior?
EAC: we have implemented the missing information
Lines 364 – Which were these conserved differences?
EAC: we have implemented the missing information
Lines 365-368 - This has no context. Where was this analysis performed? Which microalgae? Was this a part of the previous paragraph? Was this a result of a different type of analysis? It is not clear.
EAC: we have significantly rearranged and shortened this whole chapter. We believe now it reads better.
Line 373 – COX, PTGES and PTGFS without italics
EAC: done
Line 377 – Lacks a reference.
EAC: this text was removed
Line 380 – COX without italics.
EAC: done
Lines 394-398 - Rephrase and expand in other to better understand the possibility of non-enzymatic oxylipins. Why are they more advantageous than enzymatic? Is it yields of production wise?
EAC: this paragraph was moved to the conclusions section
Line 400 - What is the relevance of editing TAG synthesis? Are TAGs bioactive lipids themselves? Are they storage of bioactive lipids? What is their relevance in the context of this review paper?
We have rearranged the chapter on genetic engineering in diatoms by shortening the part on TAG production and only focusing on the available gene editing tools and only on studies reporting an increase in EPA and DHA.
Line 425 - CRISPR has been used in page 8 and only defined here. Please recheck all the document for acronyms and abbreviations to be defined the first time they are being used.
EAC: done
Lines 448-450 – Please use italics appropriately – to identify the names of moss and microalgae species and not on references and beginning of sentences that do not require italics.
EAC: done
Reviewer 2 Report
Comments and Suggestions for Authors
I would like to thank the authors for meticulously following the comments I posted. The article can be published in its present form.
Author Response
Thank you for your time in revising our manuscript.
Reviewer 3 Report
Comments and Suggestions for Authors
The authors have made a serious revision of their review article. Indeed, three figures were inserted into the text of the manuscript, in which biosynthesis and roles of bioactive lipids in human physiology, anti-inflammatory effects of microalgal bioactive lipids on intracellular signaling pathways and a scheme of prostaglandin metabolism in microalgae were discussed. In addition, the text of the article was carefully reworked according to the comments of other reviewers. Now the material looks more impressive and more clearly presented.
The reviewer has only minor comments left before he can recommend the manuscript for final publication:
1) there is no reference to Figure 2 in the body of the manuscript;
2) the article lacks information about the sources of funding for the work.
After eliminating these minor comments, this review can be published in the journal Marine Drugs.
Author Response
The authors have made a serious revision of their review article. Indeed, three figures were inserted into the text of the manuscript, in which biosynthesis and roles of bioactive lipids in human physiology, anti-inflammatory effects of microalgal bioactive lipids on intracellular signaling pathways and a scheme of prostaglandin metabolism in microalgae were discussed. In addition, the text of the article was carefully reworked according to the comments of other reviewers. Now the material looks more impressive and more clearly presented.
EAC: thank you for acknowledging the value of our manuscript and for your time in revising it. We have now implemented the missing figure reference in the text. No funding was used in this work.
The reviewer has only minor comments left before he can recommend the manuscript for final publication:
1) there is no reference to Figure 2 in the body of the manuscript;
2) the article lacks information about the sources of funding for the work.
After eliminating these minor comments, this review can be published in the journal Marine Drugs.
